# Environmental Performance of Regional Protected Area Network: Typological Diversity and Fragmentation of Forests

**Tatiana Chernenkova** [1,*]**, Ivan Kotlov** [2]**, Nadezhda Belyaeva** [1]**, Elena Suslova** [3] **and Natalia Lebedeva** [4]

1 Institute of Geography of the Russian Academy of Sciences, Staromonetniy Pereulok 29, 119017 Moscow, Russia
2 A.N. Severtsov Institute of Ecology and Evolution of the Russian Academy of Sciences, Leninsky Prospekt 33, 119071 Moscow, Russia
3 Faculty of Geography, Department of Biogeography, Lomonosov Moscow State University, Leninskiye Gory 1, 119991 Moscow, Russia
4 Environmental Protection Foundation "Verkhovye", Agrochimikov Str. 6, Novoivanovskoye Settlement, 143026 Odintsovo, Russia
* Correspondence: chernenkova@igras.ru; Tel.: +7-916-706-52-65

**Abstract:** Protected areas (PAs) are among the main tools for preserving biodiversity and creating an environment for the natural course of ecological processes. The identification of forest biodiversity is especially important for large metropolitan areas. An obvious problem in assessing the efficiency of the PAs network is the lack of up-to-date cartographic materials representing the typological diversity of vegetation. The aim of the paper is to identify forest biodiversity and fragmentation in the example of the Moscow region (MR)—the largest metropolis in Eastern Europe. The typological classification was carried out at a detailed hierarchical level—33 association groups (ass. gr.) considering the diversity of the land cover. A random forest algorithm was used for cartographic mapping (overall accuracy 0.59). Remote sensing (RS) data included Sentinel-2A, DEM SRTM, and PALSAR radar images. Six fragmentation metrics were calculated based on the raster map of forest typological diversity. A significant correlation between the forest diversity and PAs forest patch fragmentation metrics was noted. It has been established that the PAs proportion of the territory accounts for almost 20% only within the northernmost district and noticeably decreases to the south to 1–2%. At the same time, fragmentation noticeably increases from Northeast to Southwest. The category of PAs does not affect the state of the forest cover. Additionally, there was no direct influence of the anthropogenic factor from both local sources and a large regional source, i.e., the city of Moscow. It is shown that the average area of PAs, supporting 75% of the typological diversity of regional communities, was about 1000 ha. The results of the study suggest that there is a general lack of environmental protection measures in the region. It is recommended to increase the area of PAs, primarily for less fragmented forest patches, including indigenous forest-steppe and forest types of communities.

**Keywords:** forest biodiversity; protected areas; spatial modeling; Sentinel-2A; fragmentation metrics; Moscow region





## 1. Introduction

Protected areas (PAs) are among the main tools for preserving species and ecosystem diversity and creating an environment for the natural course of ecological processes [1,2]. The efficiency of PAs is assessed by a number of indicators of the ecosystem and species diversity, including the proportion of PAs in the area of the region, the average size of PAs, the area of key ecosystems, fragmentation, changes in the status of rare species, the distribution of rare species and invasive species, as well as the degree of influence on the social environment of people, etc. [3–6].

The identification of the spatial structure of plant communities is important for comparing the efficiency of positive and sustained long-term outcomes and for the in situ

conservation of biodiversity, including the ability of PAs to support ecological processes that go beyond their borders [7,8]. We understand the efficiency of PAs as an ability of a protected area to represent the characteristic properties of the surrounding area.

Many of these efficiency indicators are successfully determined using remote sensing (RS) data [9–12]. However, the available global tools allow assessing only general spatial and temporal characteristics of the vegetation cover [13], for example, the area of undisturbed forests [14]. At the same time, many studies of PAs deal with changing land use and land cover (LULC) providing the assessment of the area and the ratio of different categories of land and types of vegetation cover [15–18]. Spatial and temporal changes in the land cover structure are often estimated by comparing different plant photosynthetic activities using the normalized difference vegetation index (NDVI) [19–22]; however, this approach is not sufficient to assess the ecosystem diversity.

RS data have wider applications for obtaining more detailed data on the phytocoenotic diversity of vegetation cover and the dynamics of vegetation types [23]. First, RS data can help determine the optimal size and configuration of potential PAs for the successful functioning of species and ecosystem processes [24]. Secondly, remote sensing contributes to the baseline assessment and monitoring of the vegetation in PAs, and the location of invasive species, including the areas of anthropogenic interference [9]. Third, RS data allow identifying the degree of land cover fragmentation associated with varying degrees of anthropogenic transformation, as well as the state of ecological cores and corridors providing the supporting functions of ecosystems [25].

A promising direction for the in-depth measurement of the efficiency of PAs is the assessment of the typological diversity and dynamics of land cover [23]. Cartographic models of typological units are developed based on the RS data. Forest typological diversity in combination with fragmentation is an effective way to measure the biodiversity of forest green infrastructure (FGI) in clear terms, as well as to organize monitoring and landscape planning in urban areas [26]. It is important to emphasize that the majority of PAs suffer from the lack of a unified system for monitoring the state of FGI [27]. In this regard, RS data allow for creating baseline maps and further systematic monitoring of FGI [28,29]. The use of ensemble methods of supervised classification makes it possible to achieve considerably higher convergence of the cartographic models of typological units [30–32].

For the Moscow Region (MR), the largest metropolis in Eastern Europe, the role of PAs is extremely important as a source of maintaining ecosystem services and preserving the species richness of flora and fauna. The anthropogenic impact on the forest cover is closely related to the historical development and features of agriculture in the region. The coniferous–broad-leaved forests of the Russian Plain began to be massively replaced by arable lands as early as the 11th–12th centuries and, by the beginning of the 16th century, "the level of plowing has approached the maximum possible" [33], p. 228. After the abolition of serfdom in the 19th century, forests began to be cut down even more intensively, and high fragmentation of the landscape was formed. As a result, at the beginning of the 20th century, the forest cover of the Moscow region was only 26% [34]. Changes in agricultural practices, the creation of spruce and pine forest cultures in the 20th century, and the natural regeneration of forests on abandoned arable lands have increased the forest cover of the region up to 48% now [32].

In recent decades, the action of multidirectional processes has been observed. On the one hand, the proportion of agricultural land in the region has decreased, and there is an active overgrowth of abandoned agricultural land [35]. On the other hand, the region has seen the development of urban infrastructure, and the recreational use of natural areas, often accompanied by the withdrawal or strong transformation of forest areas. At the same time, there is an obvious lack of information on the management of PAs, particularly the lack of up-to-date cartographic materials on the typological diversity of vegetation.

According to the Ministry of Ecology and Nature Management of the Moscow Region [36], the area of PAs is about 6% of the territory of the region. How effectively does the existing network of PAs reflect the biodiversity of the region? The objective of the

paper was to identify the typological composition and fragmentation of FGI in PAs in order to maintain the biodiversity of vegetation cover around the largest metropolis in Eastern Europe (Moscow region, Russia). The following tasks were solved:

- Assessment of the composition and spatial distribution of forest cover in PAs;
- Assessment of the structure (fragmentation) of forest cover in PAs;
- Identification of the main factors that determine the composition and structure of PAs forests, taking into account natural conditions and anthropogenic pressure.

The research methodology is intended for the regular monitoring and optimization of FGI based on the network of PAs. The results of the work provide information about the current state of forest cover in PAs within different botanical districts (BDs) of the MR. In case of insufficient representation in terms of area and diversity of forests, as well as high fragmentation, recommendations are given for the improvement of the PAs network.

## 2. Materials and Methods

### 2.1. Study Area

The MR is located in the Central part of the East European (Russian) Plain—35°10′–40°15′E, 54°12′–56°55′N, and its area is 4.7 million ha. The relief of the territory is gently hilly, elevation varies from 90 to 320, on average 174 m a.s.l., and the average slope is 2.06° (0–30.9°). In accordance with the geobotanical zoning, the study area is located mainly within the zone of coniferous-deciduous forests [37]. The forest cover in the MR is represented by a succession mosaic of forests of different compositions, ages, and origins. Coniferous and mixed communities account for a significant proportion of artificial forests [38].

Six botanical districts (BDs) have been identified based on botanical zoning of the MR: 1—Lotoshinsko-Taldomsky (LT), 2—Mozhaisk-Zagorsky (MZ), 3—Noginsko-Shatursky (NSh), 4—Podolsko-Kolomensky (PK), 5—Kashirsko-Zaraisky (KZ), 6—Serebryanoprudsky (S) [39] (Figure 1).

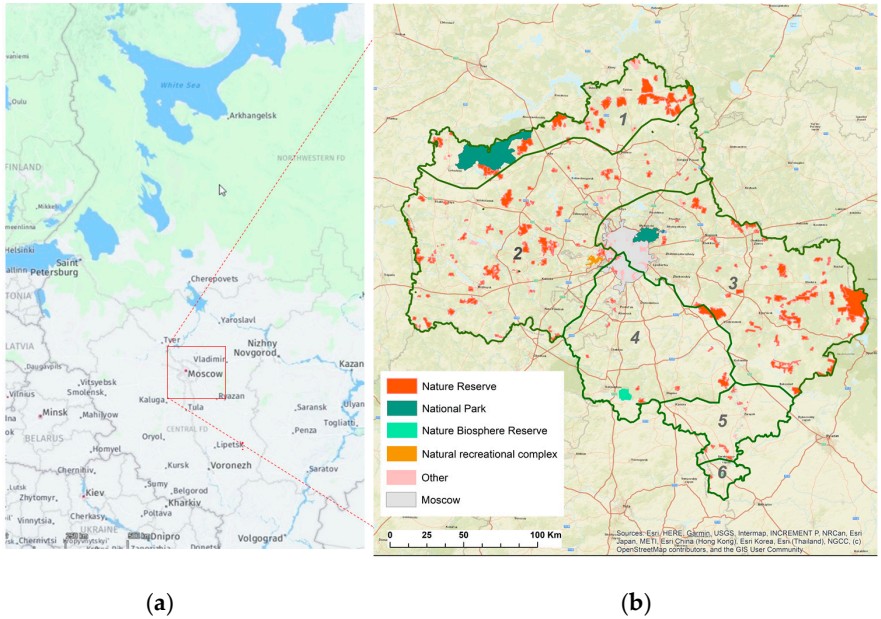

(**a**)                                                                 (**b**)

**Figure 1.** Study area (**a**) and the scheme of PAs in the MR (**b**) BDs: 1—LT, 2—MZ, 3—NS, 4—PK, 5—KZ, 6—S.

LT (#1) is characterized by the dominance of coniferous forests without broad-leaved trees and oak forest elements and a significant distribution of wetland forests and wetlands. Spruce forests are widespread here, and there are fewer pine forests, as well as pine-spruce forests. The characteristic features of MZ (#2) are the predominance of relatively eutrophic spruce forests, the presence of oak-spruce and oak forests on elevations, the minor development of wetlands, and the low presence of pine. NSh (#3) is characterized

by the predominance of pine forests (often with spruce), mostly relatively oligotrophic, and a wide distribution of peat wetlands. The vegetation of the district is similar to LT (#1) but differs from it in a much smaller distribution of spruce forests and a greater distribution of broadleaf forests. PK (#4) includes primary forests (almost exclusively broad-leaved, i.e., linden and linden-oak forests with an admixture of spruce) typical for the northern part of the broad-leaved forest subzone. In the southeastern part of the district, broad-leaved forests are completely devoid of spruce. KZ (#5) is located on the northern slopes of the Central Russian Upland and occupies almost the entire part of the MR south of the Oka River. The dominant primary vegetation of the interfluves is oak forests, typical for the southern part of the subzone of broad-leaved forests (with ash and maple, but without spruce). There are very few forests left here. Sd (#6) is a small area located in the extreme south of the MR. The primary vegetation here is forest-steppe (alternating sections of oak forests and northern steppes). At present, the territory is almost continuous agricultural land.

In general, primary (undisturbed) forests have not been preserved in the MR. At the same time, there are nominally primary forests there, which are customarily understood as forests that are close to indigenous analogs in terms of the composition of the tree and subordinate layers but differ significantly from them in terms of the age structure of forest stands.

### 2.2. History and Structure of PAs Network in the MR

The creation of PAs in the MR began around the 1930s as a part of the overall development of the nature reserve system of the Soviet Union. Initially, they were focused only on protecting the most valuable ecosystems. In the 1960s–1990s, the network of PAs in the region intensively grew, and most of them were established. The goals for establishing PAs have gradually changed from the conservation of model objects, sights and resources, through the conservation of biodiversity in the 1970s, to the conservation of landscape structure in the 1980s and 1990s [40]. Over the PAst decade, a new process of formation of additional reserves has taken place—not only unique but also many typical ecosystems have been added to the PAs. Thus, an integrated ecosystem approach is currently being used to create a comprehensive representative system of PAs.

As of 2022, there are four PAs of Federal significance, and 270 PAs of Regional significance in the MR. Among them, there are 177 Nature Reserves, 87 Natural Monuments, five Natural and recreational zones, and one Strictly protected water object. In addition, there are more than a hundred diverse PAs of local importance, mostly very small in size. The total number of PAs of different categories and levels of significance is about 400, and their total area is at least 340 thousand hectares (Table 1).

**Table 1.** Number and area of PAs in the MR.

| Category (IUCN) | Number | Total Area, ha |
|---|---|---|
| Nature Biosphere Reserve (I) [1] | 1 | 4957 |
| National Park (II) | 2 | 71,796 |
| Natural Reserve (IV) | 182 | 237,061 |
| Natural Monument (III) | 164 | 10,305 |
| Coastal recreation area (V) | 5 | 7254 |
| Strictly protected water object (V) | 1 | 7658 |
| Regional Natural Reserve (V) | 7 | 21.97 |
| Natural recreational complex (V) | 5 | 28.66 |
| Dendrological park and Botanical Garden (V) | 2 | 0.6 |
| Natural History Park (V) | 14 | 1.77 |
| Other | >100 | >4000 |
| Total | >380 | > |

[1] In brackets—IUCN category compliance [41].

### 2.3. Study Design

The typological diversity of forest cover was assessed by cartographic modeling of forest communities at the rank of association groups (ass. gr.) identified during the processing of field data. RS and field data were integrated to identify the diversity and fragmentation of the vegetation cover of PAs. Such integration was previously partially tested in the forests of the MR [32,42]. An overview of the study design is illustrated in Figure 2. The approach was divided into seven steps: (1) Field data collection and classification; (2) Processing RS and original cartographic data; (3) Modeling of forest biodiversity and mapping; (4) Analysis of the forest biodiversity of PAs in the BDs; (5) Analysis of fragmentation of PAs in the BDs; (6) Assessment of environmental factors and parameters of PAs; (7) Analysis of the main factors and protection status of PAs.

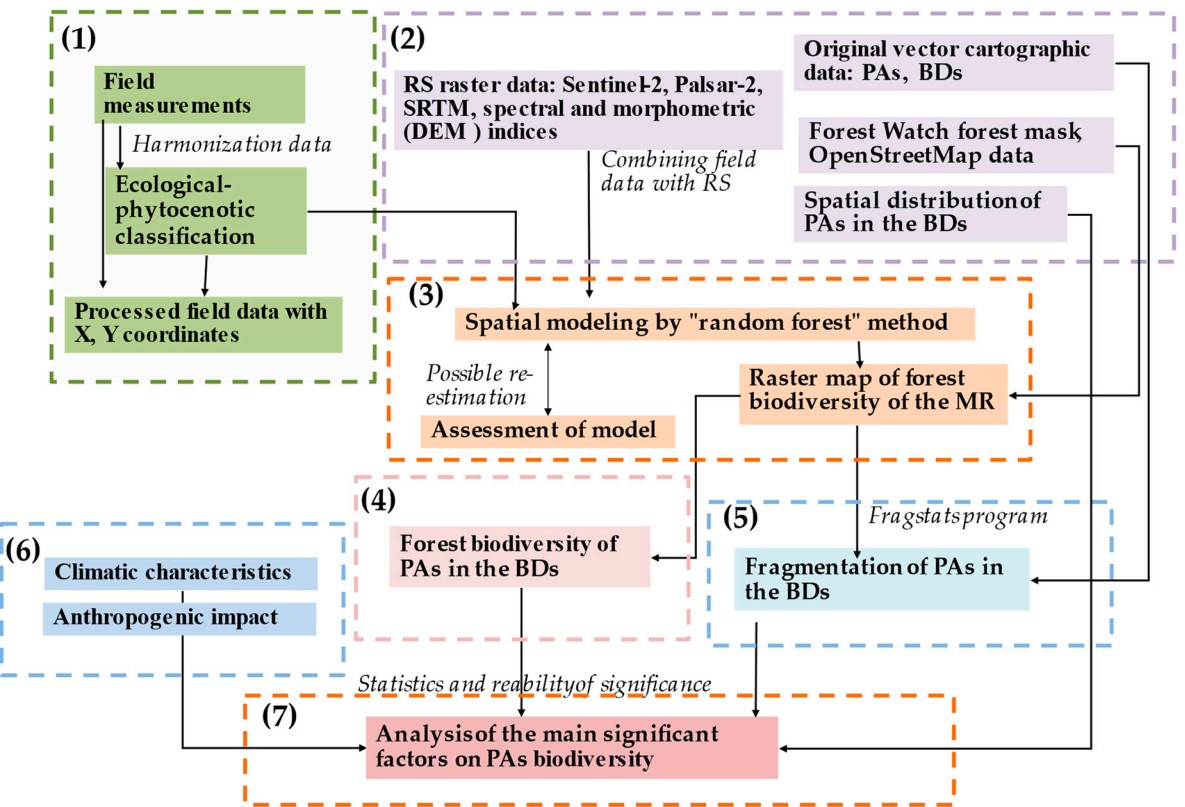

**Figure 2.** Proposed forest diversity assessment framework for PAs.

To correctly assess the state of communities close to the natural conditions of functioning, the PAs were filtered according to several criteria:

1.  The territory of the historical part of Moscow within the boundaries until 2012 is excluded from the total area of the MR. Accordingly, such categories of PAs as the Dendrological park and Botanical Garden and the Natural Historical Park were not included in the analysis.
2.  The protection of forest areas that characterize terrestrial vegetation covers most of the entire list of PAs. In this regard, Strict protected water object was excluded from consideration.
3.  PAs have various shapes and sizes and are distinguished by a variety of management systems and origins. To analyze the typical combination of types of plant communities, PAs were filtered by the minimum area. The previous results of our research allowed us to reveal the average size of a forest patch in the near zone of Moscow (14 ± 4 ha) and on the periphery (37 ± 26 ha) (unpublished data). Therefore, PAs with an area of less than 36 ha, equaling 10 × 10 pixels with a resolution of 60 m (Sentinel-2 resolution) were excluded from the analysis. Most of the small PAs are mainly represented by the



Natural Monument category or local PAs. As a result, the analyzed PAs account for 5.8% of the study area (265.2 thousand ha).

The following factors potentially determining the diversity of community types in PAs were considered:

- Area of PAs;
- Climate variables;
- Establishment date of PAs (existence time of PAs);
- Percentage of forests;
- Fragmentation;
- Category of PAs;
- Anthropogenic pressure (distance from Moscow and other large settlements).

A comparative analysis of the composition and structure of the vegetation cover within the PAs and in the reference areas (BDs) was performed to identify the efficiency of the PAs. BDs are characterized by uniform vegetation and soil cover [39].

### 2.4. Field Data and Classification

Field data include 1684 geobotanical descriptions [43]. The ecological-phytocenotic method was applied to identify ass. gr. [44]. This method has several advantages: (1) good correspondence between typological and mapped units; (2) compliance with the Russian units of forest typology; (3) hierarchical approach; (4) equal accounting for typical, rare and secondary types of forest communities, which is important from the conservation point of view.

The proportion of natural (primary) and derived communities is an important indicator of the state of forests. When assessing the successional status of forests, we distinguished the following categories: Natural forests and Derivative forests. Natural forests are formed by native tree species, developed under weak human impact and without catastrophic natural factors for a period comparable to the limiting biological age of these species. Derivative forests are serial communities after fires, anthropogenic disturbances, reclamation, and silviculture. A distinction is made between short-term derivative forests and long-term derivative forests. The recovery time for short-term derivative forests in temperate forests is 50–70 years and it is much longer—more than 100 years—for long-term derivative forests.

### 2.5. RS Data, Cartographic Modeling, and Fragmentation Evaluation

The RS data included cloud-free multispectral Sentinel-2 imagery taken over two days (20 and 23 June 2021) and processed into a seamless 11-band mosaic. In addition to 11 spectral bands, 41 spectral indices were calculated, including indices estimated as sensitive to vegetation stress [45]. SRTM digital elevation model and 10 morphometric indices were used. Two layers of Palsar-2 radar data were also used—HH and HV polarization [46]. A total of 63 raster layers were obtained.

To remove the autocorrelation, highly correlated layers along the 0.5 threshold were removed [47]. Seven raster layers are left: blue (2) and red edge (6) bands, NDWI2, BNDWI and GLI indices, absolute height and HH polarization (Table A1).

The "random forest" method was used for modeling the spatial structure of forest cover [48–50]. Orfeo Toolbox software was used [51]; 30% test sample was used for model calibration (random stratified selection) [52–54]. The fraction of points in the test sample, for which belonging to the modeled type was correctly determined, is class (formation or ass. gr.) accuracy; the total fraction of the correctly determined classes is total accuracy. The obtained maps are harmonized with the Global Forest Watch forest mask [55]. Layers of farmland, water bodies and settlements were prepared using the OpenStreetMap data [56]. Individual forest patches were extracted from the cartographic model and fragmentation metrics were calculated using Fragstats [57]. The metrics with pairwise correlations greater than 0.5 were removed [58]. Six metrics were used for further analysis: patch density (PD), largest patch index (LPI), perimeter-area fractal dimension (PAFRAC), total edge contrast index (TECI), interspersion/juxtaposition index (IJI), and splitting index (SPLIT) (Table A2).

### 2.6. Assessment of the Environmental Variables of PAs

Worldclim climatic variables were taken to assess the influence of environmental factors (spatial resolution $1 \times 1$ km) [59]. Four of the 48 variables were left after autocorrelation removal: mean monthly temperatures in January and March (T_avg jan, T_avg march), precipitation in February and April (P_avg feb, P_avg april).

The ***anthropogenic impact*** factor was estimated using remote information on the nighttime light of the Earth's surface according to the VIIRS satellite data (VNP46A3/VJ146A3 Monthly and VNP46A4/VJ146A4 Yearly Moonlight-adjusted Nighttime Lights (NTL) Product) [60]. The nighttime light correlates well with the consumption of primary energy resources at the regional level [61]. It is assumed that Nighttime lights mark several anthropogenic pressure parameters, such as population density, recreational load, and atmospheric pollution. The night radiance parameter ($W \cdot cm^{-2} \cdot sr^{-1}$) was used.

### 2.7. Analysis of Correlation between the Main Factors and Forest Biodiversity of PAs

The correlation between the typological diversity of PAs and external variables (fragmentation metrics, remoteness, illumination, and climate) was studied [62] The most significant correlations were selected (significance level $p < 0.0005$) and their physical meaning was described.

## 3. Results

### 3.1. Analysis of the Spatial Distribution of PAs

The distribution of PAs by area and by category within natural territorial complexes (BDs) is uneven. Their spatial proportion decreases from north to south and significantly correlates with the total forest cover (Figure 3a). In general, the vegetation cover is heterogeneous in composition; the forest cover of the territory decreases from north to south from 50 to 10%. It is possible to conditionally single out three BDs in the MR, which are best provided with PAs—LT, MZ and NSh (#1–3). The least prosperous part of the region is its southern part.

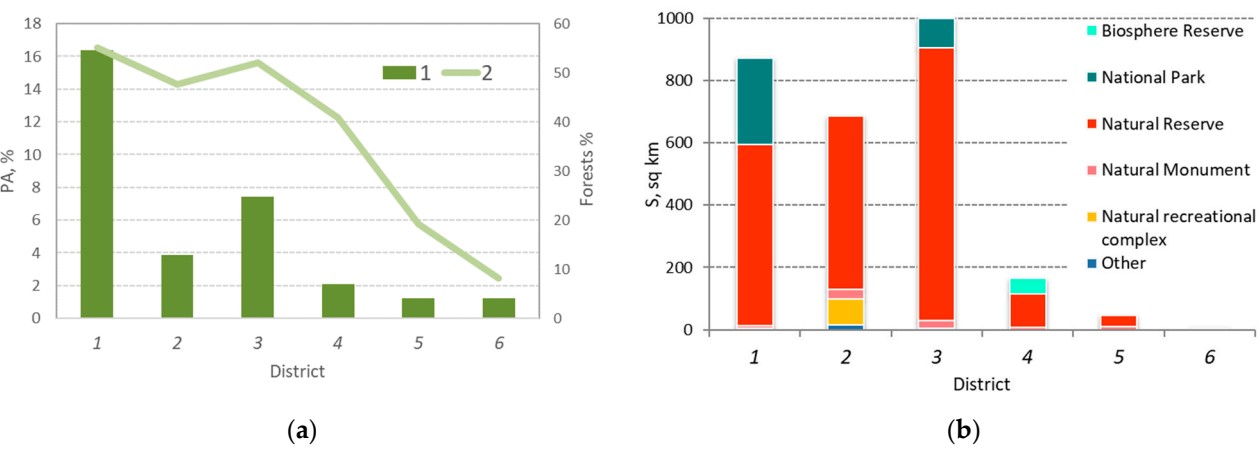

**(a)** **(b)**

**Figure 3.** Distribution of PAs in different BDs of the MR: 1—LT, 2—MZ, 3—NSh, 4—PK, 5—KZ, 6—S. (**a**)—percentage of PAs (1), forest cover (2); (**b**)—distribution of the main categories of PAs by area (km$^2$).

Natural reserves have the largest areas and highest number of PAs in the MR (Table 1, Figure 3b).

Considering the number and area of PAs by BDs, the LT district (#1) has the most nature protection measures (Table 2). Despite the relatively small number of PAs (34), their percentage is the largest there (19% of the total area of the district), while the average area of PAs is also the maximum (almost 2500 ha). The second richest district is NSh (#3). There are many PAs in the district (70), their proportion is about 8%, and the average area is about 1400 ha. Despite the fact that the number of PAs in the MZ district (#2) is maximum (97),

their average area and proportion are small (711 ha and 4%). The PAs availability in the three remaining districts, i.e., PK, KZ and S (#4–6), is relatively low. The number of PAs varies from 6 to 18 per district, they account for 1 to 2% of the district area, and the average area of PAs varies from 100 to 900 ha.

**Table 2.** Average area and number of PAs in BDs.

| Indicator | LT#1 | MZ#2 | NSh#3 | PK#4 | KZ#5 | S#6 |
|---|---|---|---|---|---|---|
| BD area, thousand ha | 431.4 | 1670.7 | 1232.1 | 795.9 | 274.8 | 49.9 |
| Forest area of the district, thousand ha | 286.8 | 849.7 | 668.2 | 328.4 | 50.7 | 3.3 |
| Area of PAs within the BD, thousand ha | 82.7 | 67.6 | 97.4 | 16.2 | 35.1 | 0.6 |
| Number of PAs | 34 | 95 | 70 | 18 | 14 | 6 |
| PAs percentage of the district area, % | 19.17 | 4.04 | 7.90 | 2.03 | 1.28 | 1.22 |
| Average area of PAs, ha | 2432.39 | 711.34 | 1371.32 | 897.84 | 251.05 | 101.10 |

*3.2. Results of Ecological-Phytocenotic Classification*

Typological units in the rank of association groups were identified as a result of the classification of forest communities. The description of 33 ass. gr. is in Table A3. A syntaxa detailed description in terms of their composition and structure, origin, as well as the dependence of forest community types on ecotope conditions is set out in a previous work [63]. The names of association groups are supplemented by the main types characterizing them. The legend reflects the successional state of forests, marked as primary (N), short (ShD)—and long-term derivative (LD) communities.

Most of the selected communities have a derivative status and are mainly represented by a short dynamic type (Table A3). Some spruce and pine forests of the boreal group (1 and 2), pine and birch forests of the hydromorphic group (18 and 28), as well as broad-leaved and spruce-broad-leaved forests (19–21) belong to the group of nemoral-type communities, similar in their composition to indigenous zonal forests. The group of alder forests with Gray alder and Black alder are also mostly of natural origin (31–33). A minor part of small-leaved forests with a long succession cycle is represented by small-leaved communities of birch with spruce and aspen (22–25 and 27).

*3.3. Cartographic Mapping of Forest Typological Diversity*

The classification results are the basis of the legend. The legend includes 41 categories, of which 1–33 ass. gr. of forest, 34—small-leaved scrubs, 35—cuts, 36—meadows, 37—open marshy habitats, 38—willow stands, 39—agricultural fields, 40—water objects, 41—settlements. The overall accuracy of the cartographic mapping of ass. gr. was 0.59. The accuracy of ass. gr. modeling is shown in Table A4. Accuracy varies from 13 to 100%. The lowest accuracies are observed for 7 ass. gr. (Sp-As/B_ShBh)—13%; 4 (Sp_Bh)—18%; 8 (Sp-As/B_Bh)—21%; 3 (Sp_ShBh)—23%; 5 (Sp-As/B_DshShG)—30%; 31 (G-Al_ MihBh)—36%. The greatest uncertainty was observed when separating spruce, Sp_Sh, Sp_ShBh ass. gr., B_Bh and aspen As_Bh, and B_MihBh and As_MihBh ass. gr. from each other. This can be explained by the difference in species composition of the tree layer and the uniform species composition of the ground layer of these communities. For this reason, we had to combine the ass. gr. in pairs: 14 (P_Sh) and 15 (P_ShBh), and 22 (B_Sh) and 23 (B_ShBh). Figure 4 shows a fragment of the vegetation map for PAs in the western sector of the region.

*3.4. Analysis of the Spatial Distribution of PAs by Forest Composition*

The distribution of community types within PAs and outside PAs—within the boundaries of BDs, was compared. Figure 5 shows different percentages of the forest types of communities (#1–33) within PAs both for individual BDs and as compared between the six districts (Figure 2a).

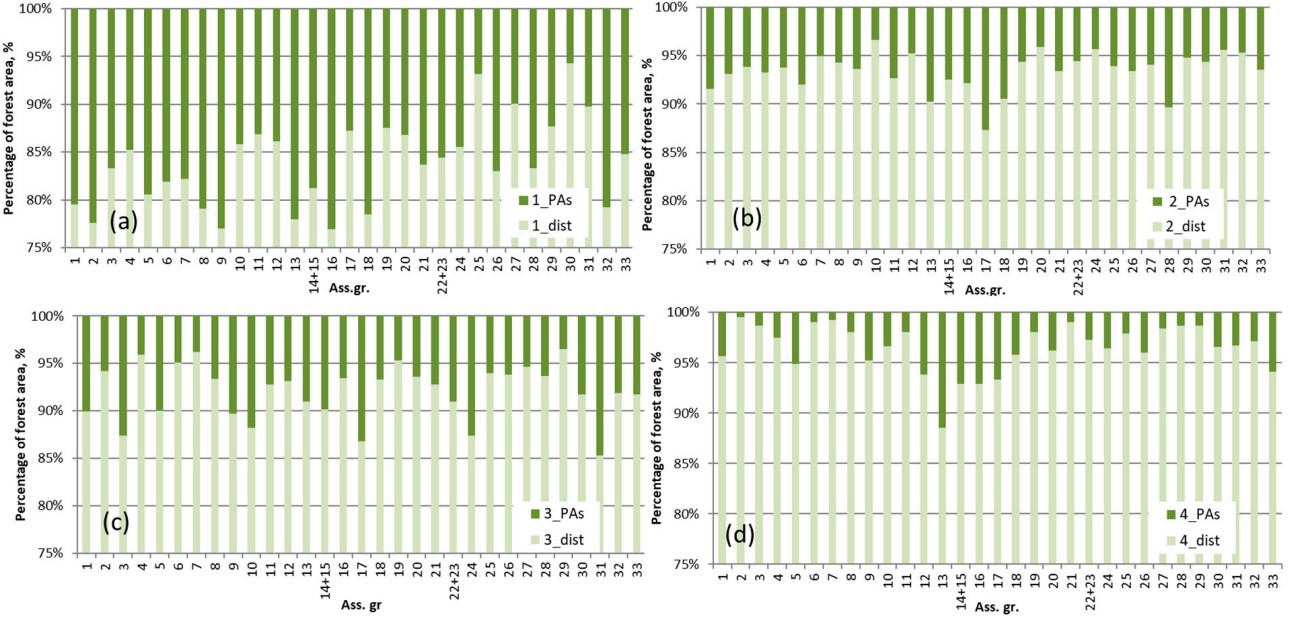

**Figure 4.** Fragment of the forest biodiversity map of PAs in the MR.

**Figure 5.** *Cont.*

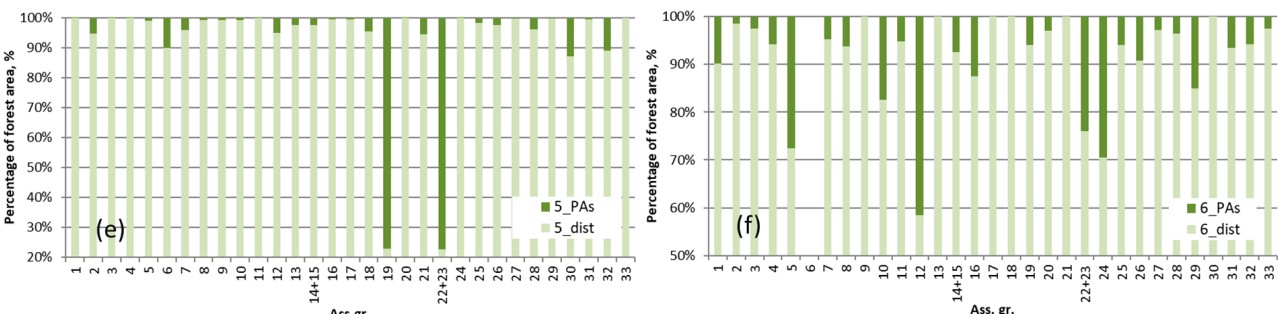

**Figure 5.** Percentage of forest communities within PAs and the BDs (km²): (**a**)—LT, (**b**)—MZ, (**c**)—NS, (**d**)—PK, (**e**)—KZ, (**f**)—S. Types of communities: 1—Sp_DshShG; 2—Sp_Sh; 3—Sp_ShBh; 4—Sp_Bh; 5—Sp-As/B_DshShG; 6—Sp-As/B_Sh; 7—Sp-As/B_ShBh; 8—Sp-As/B_Bh; 9—Sp-As/B_DshShG; 10—Sp-As/B_Sh; 11—Sp-As/B_ShBh; 12—Sp-As/B_Bh; 13—P_DshShG; 14 + 15—P_Sh; 16—P_Bh; 17—P_Mh; 18—P_DshHSh; 19—O_Bh; 20—L_Bh; 21—Bl-Sp_Bh; 22 + 23—B_Sh; 24—B_Bh; 25—B_MihBh; 26—B_Gm; 27—B_Mh; 28—B_DshHSh; 29—As_Bh; 30—As_MihBh; 31—GAl_MihBh; 32—BAl_MihBh; 33—BAl_Gm.

An analysis of the distribution of forest communities in the context of BDs showed that their maximum representation in terms of typological diversity is observed in the LT (#1) BD. In the sublatitudinal direction from district #2 to district #4, while maintaining an almost complete range of community types, their proportion is violated, and subnemoral and nonmoral types begin to predominate. In the southern part of the region (#5 and #6 districts), there is an incomplete spectrum of communities with a predominance of nemoral types. The proportion of their participation from the area of the BD is small.

The analysis of natural and derived communities showed that only district #1 has approximately equal areas of PAs forests of different succession statuses, which indicates a stable state of forest cover. There are just minor areas of natural communities and an obvious predominance of long-term derivative communities in districts #3–6, while there is a slight predominance of short-term derivative communities in district #2 (Figure 6).

The distribution of forest typological diversity by categories of PAs was assessed using the group statistics (Table 3). Two categories of PAs—Nature Biosphere Reserves and National Park account for the highest diversity of forest typological types (the average number of community types is 31). High typological diversity was also a reliable result—more than 20 community types—for the Nature Reserves and Regional Nature Reserves. For other PA categories, the forest diversity is less than 20 community types.

**Table 3.** Breakdown table of groups statistics of the typological diversity of forests by PAs categories (ANOVA: F = 6.78, *p* = 0.000001).

| PAs Category | Average Number of Ass. Gr. | Number of PAs | Std. Dev. |
|---|---|---|---|
| Nature Biosphere Reserve | 31.00 | 1.00 | |
| National Park | 30.50 | 2.00 | 0.71 |
| Natural Reserve | 21.04 | 156.00 | 6.68 |
| Natural Monument | 14.02 | 44.00 | 6.54 |
| Coastal recreation area | 17.7 | 3.00 | 12.74 |
| Regional Natural Reserve | 20.5 | 6.00 | 3.76 |
| Natural recreational complex | 17.3 | 3.00 | 10.07 |
| For all categories | 19.6 | 215.00 | 7.22 |

*3.5. Analysis of the Spatial Distribution of PAs Fragmentation Metrics*

The average area of the forest patch for the analyzed PAs within different BDs varies from 3.7 to 60.3 ha. In BDs #1–4 the indicator differs insignificantly between the district and the PAs. The average area of the PAs forest patch in the KZ (#5) district is much larger

than that in the whole district (60 ha vs. 10 ha), while, on the contrary, it is smaller in S (#6) district (4 ha vs. 8 ha).

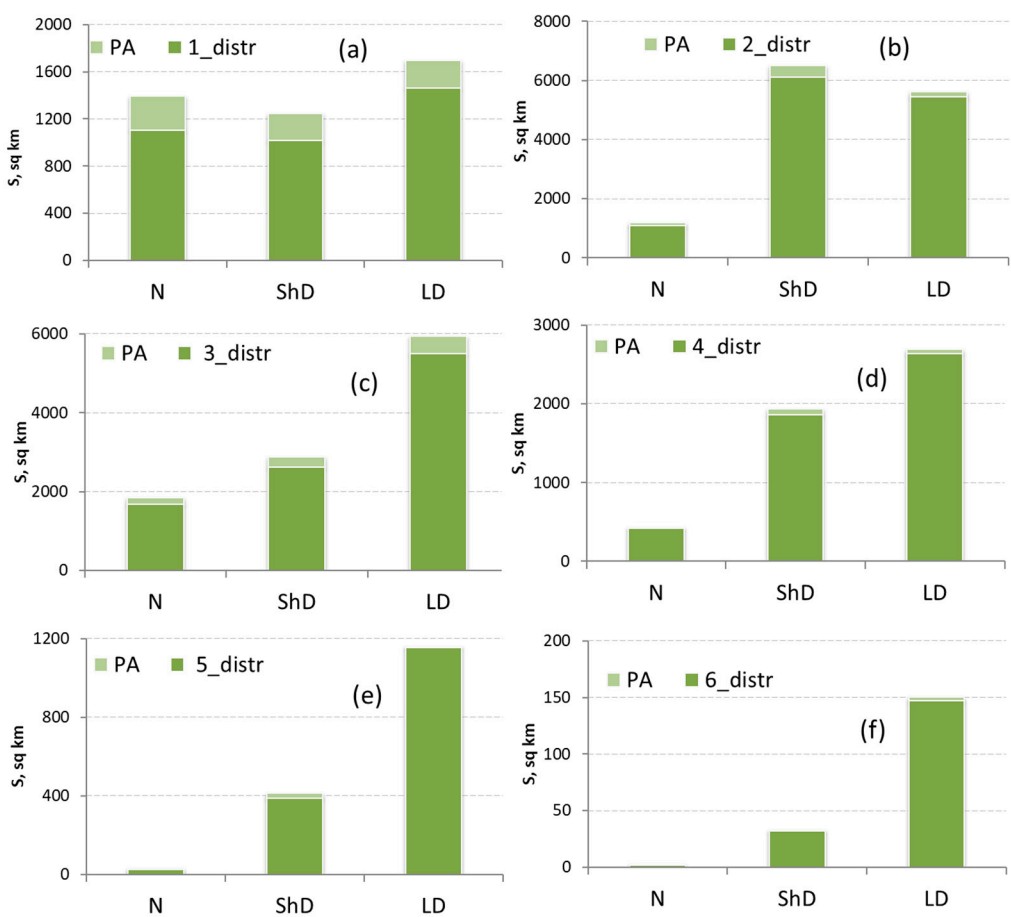

**Figure 6.** The area of forest communities with different successional statuses within the protected areas (PA) and the district as a whole (distr) (km$^2$). Successional status: 1—natural forests, 2—short-term derivative forests, 3—long-term derivative forests. BDs: (**a**)—LT, (**b**)—MZ, (**c**)—NS, (**d**)—PK, (**e**)—KZ, (**f**)—S.

The average patch density (PD) in geobotanical districts decreases from about 3.5 to 0.8 ha from north to south. At the same time, the average patch density for PAs is much higher in Districts #1–5 (from 7.6 to 16.87) and drops sharply to 3.17 ha in District #6 (Table 4).

The largest patch index (LPI) for BDs is approximately the same (0.4–2.8%); the lowest LPI values are typical for Districts #2 and 5 (0.6 and 0.4%, respectively), and the highest is in District #1 (2.8). The LPI index for PAs exceeds the indicator for a corresponding district; it amounts to 38–57% for PAs in districts #1–5 and drops sharply to 6% for PAs in district #6.

The fractal dimension index varies within a small range (1.39–1.55). The fractal dimension for the district is higher than for PAs in all districts except #6. The contrast index is approximately the same for districts #1–4 (13.5–18.2%) and drops sharply for districts #5 and 6 (7.2% and 1.6%). The contrast ratio of PAs is slightly lower (9.7–14.2%) for districts #1–4 and 4.8% and 2.8% for districts #5 and 6, respectively. The mixing index (IJI) is poorly differentiated both by districts as a whole and by PAs (49–67%). Districts #1–4 are generally slightly more mixed than their PAs. Moreover, the situation is reversed for Districts #5 and 6.

### 3.6. Assessment of the Main Environmental Variables and Parameters of PAs

Environmental variables were prepared to identify the most significant factors that determine the forest typological diversity in PAs. Table 5 shows pairwise correlations between various parameters of PAs: area, perimeter, fragmentation metrics, climatic variables, light index (anthropogenic impact), and date of PAs establishment. There is a significant correlation of 0.76 between area and perimeter. This is a common relationship, so we remove the perimeter from further analysis. Another interesting phenomenon is the −0.68 correlation between the distance to Moscow and precipitation in February. Probably, this may be due to the warming effect of the city, which is especially pronounced in winter, so we remove the factor of February precipitation from further analysis. Additionally, a correlation of −0.61 is between the distance to Moscow and the light index, which is quite logical, and the light index is removed from further analysis. As a result, highly correlated spatial and temporal variables were removed, and independent significant factors were left.

**Table 4.** Forest fragmentation metrics in PAs and in BDs.

| Indicators, Units | | LT#1 | MZ#2 | NSh#3 | PK#4 | KZ#5 | S#6 |
|---|---|---|---|---|---|---|---|
| Average patch area, ha | BD | 16.31 | 13.34 | 15.27 | 16.54 | 10.10 | 8.05 |
| | PAs | 17.88 | 12.89 | 12.54 | 14.50 | 60.33 | 3.73 |
| PD, n/100 ha | BD | 3.17 | 3.55 | 3.14 | 2.41 | 1.71 | 0.78 |
| | PAs | 16.87 | 14.59 | 8.68 | 7.62 | 11.57 | 3.17 |
| LPI, % | BD | 2.80 | 0.60 | 1.54 | 1.81 | 0.44 | 1.29 |
| | PAs | 42.67 | 43.95 | 38.35 | 38.51 | 57.17 | 6.33 |
| PAFRAC, none | BD | 1.48 | 1.55 | 1.51 | 1.51 | 1.45 | 1.46 |
| | PAs | 1.41 | 1.45 | 1.44 | 1.40 | 1.39 | 1.49 |
| TECI, % | BD | 18.24 | 13.47 | 14.26 | 17.53 | 7.20 | 1.58 |
| | PAs | 14.20 | 9.72 | 11.30 | 12.20 | 4.79 | 2.75 |
| IJI,% | BD | 65.50 | 61.63 | 62.63 | 66.47 | 50.32 | 49.14 |
| | PAs | 60.08 | 60.08 | 61.54 | 57.92 | 57.88 | 58.93 |

**Table 5.** Pairwise correlations between various environmental parameters.

| | Area, ha | Distance to Moscow, km | Perimeter, km | Pavg Feb | Pavg April | Tavg Jan | Tavg March | Date of Creation PAs | NTL | PD | LPI | PAFRAC | TECI | IJI | SPLIT |
|---|---|---|---|---|---|---|---|---|---|---|---|---|---|---|---|
| | 1 | 2 | 3 | 4 | 5 | 6 | 7 | 8 | 9 | 10 | 11 | 12 | 13 | 14 | 15 |
| 1 | | 0.08 | **0.76** | −0.06 | −0.06 | −0.21 | 0.04 | −0.07 | −0.03 | −0.35 | −0.22 | 0.22 | 0.22 | −0.07 | 0.22 |
| 2 | | | 0.00 | **−0.68** | 0.35 | −0.39 | 0.03 | −0.10 | **−0.61** | −0.10 | 0.07 | 0.02 | 0.17 | −0.12 | −0.06 |
| 3 | | | | 0.02 | −0.04 | −0.11 | 0.07 | −0.07 | 0.10 | −0.39 | −0.30 | 0.25 | 0.13 | −0.06 | 0.18 |
| 4 | | | | | −0.10 | 0.29 | −0.04 | 0.06 | **0.51** | 0.05 | −0.01 | 0.00 | −0.21 | 0.13 | 0.04 |
| 5 | | | | | | −0.35 | 0.15 | −0.01 | −0.39 | 0.11 | 0.02 | 0.25 | −0.15 | −0.02 | 0.00 |
| 6 | | | | | | | −0.10 | 0.07 | 0.38 | 0.17 | −0.09 | −0.07 | −0.11 | 0.14 | 0.04 |
| 7 | | | | | | | | 0.06 | 0.16 | 0.08 | 0.07 | −0.02 | 0.06 | 0.01 | −0.10 |
| 8 | | | | | | | | | 0.13 | 0.10 | −0.08 | −0.03 | −0.06 | 0.06 | −0.03 |
| 9 | | | | | | | | | | 0.08 | −0.11 | −0.17 | −0.25 | 0.12 | 0.00 |
| 10 | | | | | | | | | | | −0.27 | −0.12 | −0.21 | 0.46 | −0.14 |
| 11 | | | | | | | | | | | | −0.22 | 0.10 | −0.28 | −0.14 |
| 12 | | | | | | | | | | | | | 0.23 | −0.11 | 0.22 |
| 13 | | | | | | | | | | | | | | −0.13 | −0.16 |
| 14 | | | | | | | | | | | | | | | −0.08 |
| 15 | | | | | | | | | | | | | | | |

Significant correlations are highlighted in red (*p* < 0.005). Correlations > 0.5 are marked with bold.

### 3.7. Analysis of Correlation between the Main Factors and the Forest Biodiversity of PAs

Selected spatial and temporal variables (environmental factors) were analyzed for their correlation with the typological diversity of PAs (Table 6). The highest significant correlations are observed for the PAFRAC fragmentation metric (0.34), as well as the area (0.32) and distance to Moscow (−0.30).

**Table 6.** Correlations of spatial and temporal factors with the typological diversity of PAs.

| Area, ha | Distance to Moscow, km | P_avg Feb | T_avg Jan | T_avg March | Date of PAs Establishment | PD | LPI | PAFRAC | TECI | IJI | SPLIT |
|---|---|---|---|---|---|---|---|---|---|---|---|
| 0.32 | −0.30 | −0.03 | 0.11 | 0.10 | 0.01 | −0.18 | −0.19 | 0.34 | 0.19 | 0.03 | −0.06 |

Significant correlations are highlighted in red ($p < 0.005$).

Figure 6 shows scatterplots and approximating functions between typological diversity and PAFRAC, TECI, PAs area, distance to Moscow and the establishment date. Table 6 and Figure 7 show a linear positive correlation between the typological diversity of PAs and the fractal dimension of PA and its patches (PAFRAC) and a logarithmic positive correlation with the area of PAs. Specifically, most of the PAs are smaller than 5 sq. km, and only a few PAs are larger than 60 sq. km. PAs with low typological diversity are mainly within non-forest areas or broad-leaved forests, characterized by a low diversity of community types. It is important that in the range of PAs area of 8–12 sq. km there is an increase in community types up to 75% of their total number.

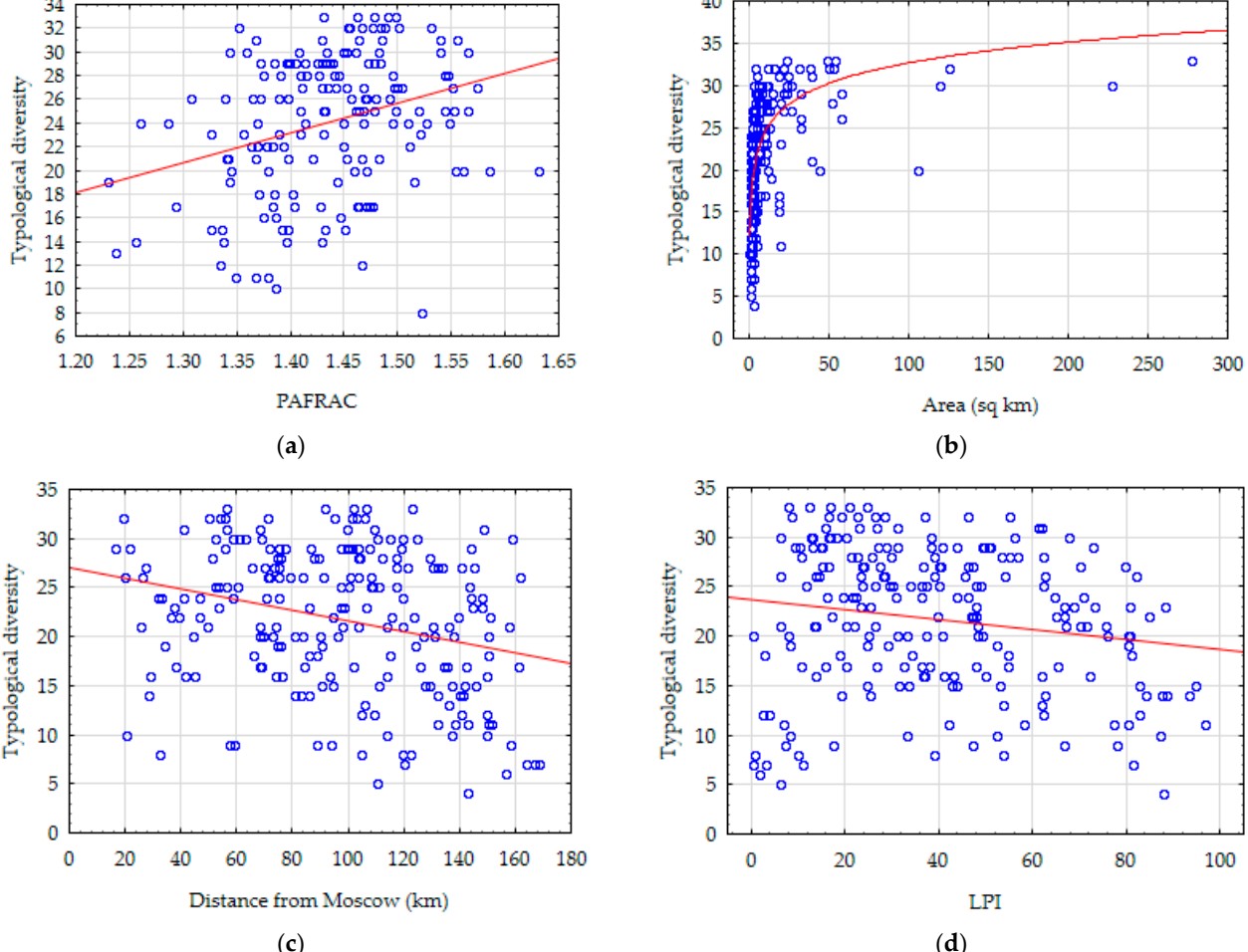

(**a**)     (**b**)

(**c**)     (**d**)

**Figure 7.** *Cont.*

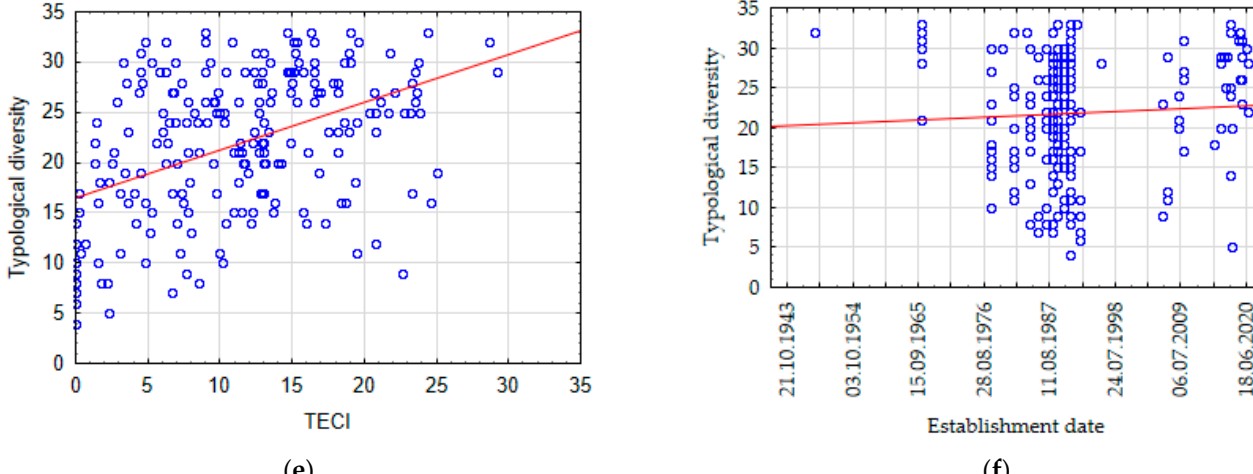

(**e**)                                                   (**f**)

**Figure 7.** Scatterplots and approximating functions between typological diversity and environmental variables: (**a**)—PAFRAC, (**b**)—area of PAs, (**c**)—distance to Moscow, (**d**)—LPI, (**e**)—TECI, (**f**)—establishment date.

The distance from Moscow shows a negative linear correlation with the typological diversity of PAs. The largest patch index (LPI) is also negatively linearly related to typological diversity. The overall contrast of the boundaries of the units included in the PAs has a positive linear correlation with typological diversity. The date of PAs establishment has no influence on the typological diversity.

## 4. Discussion

The new key target discussed under the Framework of the Convention on Biological Diversity (CBD) focuses on expanding the coverage of PAs to 30% of the Earth by 2030. It is also emphasized that not only is increasing the quantity of the world's common protected and PAs (format "30 × 30") crucial for the conservation of biodiversity but their effectiveness to achieve successful conservation results must be assessed [64]. Research has shown that about 40–50% of global PAs suffer from major deficiencies in management [65,66]. In the MR, taking into account areas joined to Moscow in 2012, the percentage of PAs is about 6% and their distribution within natural territorial complexes (BD) is uneven. There is a downward gradient from the northeast to the south and southeast. Such indicators as the number of PAs, their percentage of the area and the average area fall by one order when comparing the northernmost and southernmost districts. Geographical differences in protected area coverage are a common problem [8,11,67] and need to be addressed.

The current study proposes a methodology for the baseline assessment of the typological diversity of PAs of the MR. Thirty-three association groups were identified according to the ecological-phytocoenotic classification of field sample plots. The identified community types are characterized according to a unified scheme, taking into account the diagnostic features, composition and structure of communities, succession status, as well as the ecology of habitats. An inventory of forest diversity for protected areas of such a level of detail has not been carried out earlier in the regions.

The random forest algorithm was successfully used for a combination of remote sensing data (Sentinel-2A, DEM SRTM, PALSAR radar images) and field data. Cartographic modeling of forest cover within the framework of PAs is a continuation of previous studies for the whole region [42]. Recognition of typological units was carried out at the maximum level of hierarchy detail (overall accuracy 0.59). Similar model accuracy (0.57) was obtained in the closed canopy evergreen natural forest study in Tanzania and Kenya [68]. The relatively low accuracy of our model is explained by the complex polydominant composition of the tree layer, the combination of nemoral and boreal species groups in subordinate layers, and the silvicultural origin of most coniferous communities. Higher model accuracies [11]

are demonstrated in a number of papers using multi-spectral imagery for tree-species classification for eucalyptus plantations, by Du et al. [69] for crop area, De Alban et al. [70] for mangrove forest cover and Guirado et al. [71] for tree cover, which is explained by the simple monodominant composition of plantations. It is typical that the composition of the community types noted in the PAs (33 ass. gr. considering the diversity of the shrub and ground layers), as a whole, reproduces their entire set in the MR [32]. To improve the accuracy of the model, it is necessary (1) to improve the database of field descriptions, more evenly distributed in space and taking into account rare and remote habitats; (2) to provide the minimum number of descriptions of association groups to at least 50 (additional 494 descriptions), and in the long term, to 80 (1240 additional descriptions).

The methodological approach used to assess the biodiversity and fragmentation of PAs within the reference areas (BDs) provides additional information about the effectiveness of PAs to support ecological processes that go beyond their boundaries [7],8]. An analysis of the uneven distribution of forest communities in the BD can be explained by the botanical and geographical patterns of the distribution of forest cover and its characteristics within the region as a whole [39].

The ratio of forests of different successional statuses is an important indicator of forest stability and reproducibility. This issue is practically not presented in the scientific literature at the regional level in relation to the forest vegetation of PAs. In our study, we found that the most favorable environment (an equal ratio of primary, short- and long-term derivative forests) is in the northernmost part of the region. In the southern part of the region, a critical situation is observed—the ratio of forests of different successional statuses is uneven,—long-term derivative communities of small-leaved trees (birch and aspen) prevail. This certainly indicates the unsatisfactory state of forests according to this criterion.

Fragmentation parameters vary in different ways. The geographical differentiation of the region's territory and, as a result, the greater agricultural development of the southern part in comparison with the central and northern parts have affected the decrease in the indicators of most fragmentation metrics in the submeridional direction.

The average patch density of PAs in all districts is higher than the average patch density in the district. This is since the majority of PAs are focused on the protection of forest communities. The density of PAs also decreases along the submeridional gradient.

Based on the LPI index, one can assume the existence of a sublatitudinal fragmentation gradient that increases from east to west for all regions overall. Regarding the LPI index for PAs, there is a non-linear drop in the value of these parameters by 2.5 times from north to south. Such variability is difficult to explain by natural factors; rather, the role of a small number of PAs in districts in the southern part of the MR and, as a result, high dispersion of values, is manifested here. Noteworthy is the distribution of fractal dimension values (PAFRAC). For all districts, except for the southernmost, the fractal dimension of the PA is lower than the fractal dimension of the district. This can be interpreted from the point of view that, in terms of fractal dimension, PAs do not cover all the diversity of spatial forms available in the district.

The nature of the Contrast Index (TECI) generally shows that the northern and central parts have a relatively higher diversity of communities than one. The Contrast Index for PAs in most of the region shows that their diversity is slightly lower than in general for the respective districts; this is partly because the PAs are predominantly represented by forest communities. The PAs in the southern area, on the contrary, have a higher contrast than in the district as a whole.

The ratio of the mixing index (IJI) for the districts as a whole and for the PAs indicates that the PAs of the northern and central parts are characterized by specialization in certain types of communities, and, conversely, the PAs of the southern area are characterized by greater generalization and coverage of the entire diversity of typical regional communities.

An interesting result is the nature of the relationship between typological diversity and fragmentation metrics. Here, first, we can note a direct relationship between the complexity of the configuration of the units included in the PAs and the contrast of their boundaries

(the PAFRAC and TECI fragmentation metrics) with the general typological diversity of the PAs. This conclusion, on the one hand, directly correlates with the fact established by Aldo Leopold back in the 1930s that marginal habitats support a relatively higher abundance and diversity of species [72]. At the same time, one should not forget that much later, the negative consequences of the artificial creation of edge habitats were also revealed, including structural damage and degradation of forest stands [73]. Secondly, there is a weakly negative relationship between typological diversity and the largest division index (LPI), which may indicate that the presence of one or several large dominant divisions belonging to the same association groups suppresses the role and participation of other association groups and thus directly reduces the diversity of community types. The current study emphasizes the significance of fragmentation metrics previously shown for protected areas for example in forests in Brazil [74] of forests in China [75]. The study has sound advantages as well as it is concerned also with shape and isolation metrics unlike most studies focused on birds, and on size effects rather than isolation, and on species presence rather than population sizes [76].

The selection of proper scale (pixel size) may impact the results of fragmentation analysis but a much more significant impact is connected with the map extent of the study [77]. It means that further studies should be performed within the same map extent for correct comparison of results.

It seems quite natural that the larger the area of PAs, the greater the typological diversity represented in it. An analysis of the distribution of the typological composition by categories of PAs showed that the highest typological diversity is represented by categories with the maximum area, but single in terms of the number of PAs (Nature Biosphere Reserve and National Park). It is shown that the average area of PAs, which ensures the maintenance of 75% of the typological diversity of communities in the study region, is about 1000 ha. A similar conclusion was obtained in the work [68].

The establishment of a relationship between various parameters of PAs (area, perimeter, fragmentation metrics, climatic variables, light index, and date of PAs establishment) and forest diversity of PAs makes it possible to assess the contribution of natural and anthropogenic factors in the formation of diversity. In particular, the category of PAs does not affect the state of the forest cover. Additionally, there was no direct influence of the anthropogenic factor from both local sources and a large regional source—the city of Moscow. The negative relationship between typological diversity and distance from Moscow is not so obvious. This can be justified by the same fragmentation of the units included in the PAs. A significant relationship between the typological diversity of forests was noted with the total area of the PAs, shape complexity and boundaries contrast. It is shown that the conservation status of forests in the MR differs by groups of BDs. Near a large metropolis, there is a relatively more frequent change in land use and forest use.

## 5. Conclusions

The current study presents the method for estimating forest typological diversity, successional status, and fragmentation based on the integrated use of ground survey and remote sensing data. The proposed methodological approach made it possible to significantly expand the practice of forest biodiversity inventory and to create a digital large-scale cartographic model for PAs in the MR at a detailed typological level for the first time. Assessment of the biodiversity and fragmentation of PAs within the reference areas (BDs) allowed us to reveal their conservation status and make recommendations. It is shown that the conservation status of forests in the MR differs by groups of BDs. Most of the districts are characterized by the specialization of PAs in certain forest communities and, in general, high quantitative and area indicators, and insufficient coverage of PAs of the entire diversity of communities. It has been established that the provision with PAs only for the northernmost district is almost 20% and noticeably decreases to the south to 1–2%. At the same time, fragmentation noticeably increases from the northeast to the southwest of the MR. On the contrary, the districts in the south of the region are characterized by the

generalization of PAs and, in general, complete coverage of the community diversity, but this is since the proportion and diversity of forest cover in these districts is much lower. As a result, one can note a general lack of environmental protection measures in the region. It is recommended to increase the area of PAs, primarily for less fragmented habitats, including indigenous forest-steppe and forest types of communities.

**Author Contributions:** Conceptualization T.C.; methodology T.C. and I.K.; software I.K.; validation T.C. and I.K.; formal analysis T.C., I.K. and N.B.; conducted the fieldwork T.C., N.L., E.S. and N.B.; resources T.C. and I.K.; data curation T.C. and I.K.; writing—original draft preparation, T.C.; visualization T.C. and I.K.; supervision T.C.; project administration T.C.; funding acquisition T.C. All authors have read and agreed to the published version of the manuscript.

**Funding:** The study was conducted under state research tasks (0148-2019-0007) of the Institute of Geography RAS and Severtsov Institute of Ecology and Evolution RAS Historical ecology and biogeocenology 121122300052-5 (0089-2021-0008).

**Data Availability Statement:** Research data can be obtained from corresponding author upon request.

**Acknowledgments:** The authors express their sincere gratitude to the Environmental Fund "Verkhovye" for many years of cooperation and provided data. We thank Olga Nits for her help in clarifying the boundaries of PAs.

**Conflicts of Interest:** The authors declare no conflict of interest.

## Appendix A

**Table A1.** Spatial feature raster layers with pairwise correlations less than 0.5.

| No. | Name | Description | Characteristics, Formula |
|---|---|---|---|
| 1 | B02—Blue | Sensitivity to plant aging, carotenoids, browning and soil background; atmospheric correction (aerosol scattering) | 458–522 нм |
| 2 | B06—Red Edge | Red edge position, atmospheric correction (aerosol scattering) | 733–747 нм |
| 3 | NDWI2 | Normalized differential water index. Emphasizes the humidity of habitats [45,78] | $\frac{Green - NIR}{Green + NIR}$ |
| 4 | BNDWI | Normalized difference index of blue and infrared. Relationship with leaf area index and dry biomass volume [45,79] | $\frac{NIR - BLUE}{NIR + BLUE}$ |
| 5 | GLI | Green leaf index. Chlorophyll and leaf surface characteristics based on visible bands [45,80] | $\frac{2 \times Green - Red - Blue}{2 \times Green + Red - Blue}$ |
| 6 | DEM SRTM (elevation) | Position relative to watersheds and stream valleys [81] | meters |
| 7 | HH Palsar | Textural heterogeneity of the crown surface, tree layer height, biomass [46] | conventional units |

**Table A2.** Landscape metrics with pairwise correlations less than 0.5.

| Metrics | Formula | Units |
|---|---|---|
| PD Patch density | $PD = \frac{N}{A}(10000)(100)$ <br> $N$—number of patches in PAs <br> $A$—area of PAs | Number per 100 ha |
| LPI Largest patch index | $LPI = \frac{\max(a_{ij})}{A}(100)$ <br> $a_{ij}$—area (m$^2$) of patch $ij$ <br> $A$—area of PA | Percentage |

**Table A2.** *Cont.*

| Metrics | Formula | Units |
|---|---|---|
| PAFRAC<br>Perimeter-area fractal dimension | $$PAFRAC = \frac{\left[N\sum_{i=1}^{m}\sum_{j=1}^{n}\ln p_{ij}\ln a_{ij}\right] - \left[\left(\sum_{i=1}^{m}\sum_{j=1}^{n}\ln p_{ij}\right)\left(\sum_{i=1}^{m}\sum_{j=1}^{n}\ln a_{ij}\right)\right]}{\left(N\sum_{i=1}^{m}\sum_{j=1}^{n}\ln p_{ij}^{2}\right) - \left(\sum_{i=1}^{m}\sum_{j=1}^{n}\ln p_{ij}\right)}$$<br>$a_{ij}$—area (m$^2$) of patch $ij$<br>$p_{ij}$—perimeter (m) of patch $ij$<br>$N$—number of patches in PA | None |
| TECI<br>Total edge contrast index | $$TECI = \frac{\sum_{i=1}^{m}\sum_{k=i+1}^{m}(e_{ik}d_{ik})}{E^{*}}\,(100)$$<br>$e_{ik}$—total length of boundaries between formations $i$ and $k$ in the SPNA, including the outer boundaries of the PAs belonging to the formation $i$.<br>$E^{*}$—total length of boundaries in PAs, including outer boundaries.<br>$d_{ik}$—contrast of boundaries between formations $i$ and $k$. | Percentage |
| IJI<br>interspersion/juxtaposition index | $$IJI = \frac{-\sum_{i=1}^{m}\sum_{k=i+1}^{m}\left[\left(\frac{e_{ik}}{E}\right)\ln\left(\frac{e_{ik}}{E}\right)\right]}{\ln(0.5[m(m-1)])}\,(100)$$<br>$e_{ik}$—total length of boundaries between formations $i$ and $k$ in PAs.<br>$E$—total length of boundaries in PAs, excluding outer boundaries.<br>$m$—number of formations in PAs | Percentage |
| SPLIT<br>Splitting index | $$SPLIT = \frac{A^{2}}{\sum_{i=1}^{m}\sum_{j=1}^{n}a_{ij}^{2}}$$<br>$a_{ij}$—area (m$^2$) of patch $ij$<br>$A$—protected area | none |

**Table A3.** Forest community types.

| Formation | | Association Group | Legend Index | Type of Dynamics |
|---|---|---|---|---|
| 1 | Spruce forests (Sp)<br>(*Picea abies*) | 1 — Spruce forests with birch, aspen and pine dwarf shrubs–small herb–green moss (*Vaccinium myrtillus, V. vitis-idaea, Oxalis acetosella, Calamagrostis arundinacea, Luzula pilosa, Pleurozium schreberi, Hylocomium splendens*) | Sp_DshShG | N |
| | | 2 — Spruce forests with birch and aspen small herb (*Oxalis acetosella*) | Sp_Sh | ShD |
| | | 3 — Spruce forests with birch, aspen and pine small herb–broad herbs (*Oxalis acetosella, Carex pilosa, Galeobdolon luteum*) | Sp_ShBh | ShD |
| | | 4 — Spruce forests with birch, aspen, pine, oak and linden broad herbs (*Galeobdolon luteum, Aegopodium podagraria, Carex pilosa, Anemonoides nemorosa, Oxalis acetosella*) | Sp_Bh | ShD |
| 2 | Spruce—aspen/birch forests (Sp-As/B)<br>(*Picea abies—Populus tremula—Betula pendula, B. pubescens*) | 5 — Spruce—aspen/birch dwarf shrubs–small herbs–green mosses (*Vaccinium myrtillus, Oxalis acetosella, Calamagrostis arundinacea, Orthilia secunda, Pleurozium schreberi, Hylocomium splendens*) | Sp-As/B_DshShG | ShD |
| | | 6 — Spruce—aspen/birch small herbs (*Oxalis acetosella, Dryopteris carthusiana, Rubus saxatilis, Plagiomnium affine*) | Sp-As/B_Sh | ShD |

**Table A3.** *Cont.*

| | Formation | | Association Group | Legend Index | Type of Dynamics |
|---|---|---|---|---|---|
| | | 7 | Spruce—aspen and spruce—small herbs–broad herbs (*Corylus avellana, Athyrium filix-femina, Dryopteris carthusiana, D. filix-mas, Oxalis acetosella, Rubus saxatilis, Convallaria majalis, Galeobdolon luteum, Atrichum undulatum, Hylocomoim splendens*) | Sp-As/B_ShBh | ShD |
| | | 8 | Spruce—birch with oak and linden broad herbs (*Aegopodium podagraria, Carex pilosa, Pulmonaria obscura, Dryopteris filix-mas, Galeobdolon luteum, Eurhynchium angustirete*) | Sp-As/B_Bh | ShD |
| 3 | Pine—spruce forests (P-Sp) (*Pinus sylvestris—Picea abies*) | 9 | Pine—spruce with birch dwarf shrubs–small herbs–green mosses (*Vaccinium myrtillus, Oxalis acetosella, Dryopteris carthusisna, Pleurozium schreberi, Hylocomium splendens*) | Sp-As/B_DshShG | ShD |
| | | 10 | Pine—spruce small herbs (*Oxalis acetosella*) | Sp-As/B_Sh | ShD |
| | | 11 | Pine—spruce small herbs–broad herbs (*Corylus avellana, Oxalis acetosella, Galeobdolon luteum, Dryopteris carthusiana*) | Sp-As/B_ShBh | ShD |
| | | 12 | Pine—spruce with birch broad herbs (*Athyrium filix-femina, Galeobdolon luteum, Carex pilosa, Oxalis acetosella*) | Sp-As/B_Bh | ShD |
| 4 | Pine forests (P) (*Pinus sylvestris*) | 13 | Pine with spruce and birch dwarf shrubs–small herbs–green mosses (*Vaccinium yrtillus, V. vit1s-idaea, Pteridium aquilinum, Calamagrostis arundinacea, Convallaria majalis, Luzula pilosa, Maianthemum bifolium, Hylocomium splendens, Pleurozium schreberi, Dicranum scoparium*) | P_DshShG | N |
| | | 14 | Pine with spruce and birch small herbs (*Corylus avellana, Oxalis acetosella, Vaccinium myrtillus, Calamagrostis arundinacea*) | P_Sh | ShD |
| | | 15 | Pine with spruce and birch partly with oak and linden small herbs–broad herbs (*Oxalis acetosella, Gymnocarpium dryopteris, Galeobdolon luteum, Dryopteris carthusiana, Athyrium filix-femina, Aegopodium podagraria*) | P_ShBh | ShD |
| | | 16 | Pine with spruce, birch, oak, and linden broad herbs (*Carex pilosa, Convallaria majalis, Galeobdolon luteum, Ranunculus cassubicus, Oxalis acetosella*) | P_Bh | ShD |
| | | 17 | Pine with spruce and birch meadow herbs (*Calamagrostis arundinacea, Poa angustifolia, Convallaria majalis, Fragaria vesca*) | P_Mh | ShD |
| | | 18 | Pine with birch (*Betula pubescens*) dwarf shrubs–herbal-sphagnum (*Chamaedaphne calyculata, Ledum palustre, Vaccinium myrtillus, V. uliginosum, Oxycoccus palustris, Eriophorum vaginatum, Sphagnum angustifolium, S. magellanicum*) | P_DshHSh | N |
| 5 | Oak forests (O) (*Quercus robur*) | 19 | Oak with linden, spruce, and birch broad herbs (*Aegopodium podagraria, Carex pilosa, Galeobdolon luteum*) | O_Bh | N |
| 6 | Linden forests (L) (*Tilia cordata*) | 20 | Linden broad herbs (*Carex pilosa, Aegopodium podagraria, Mercurialis perennis, Pulmonaria obscura*) | L_Bh | N |

**Table A3.** *Cont.*

| | | Formation | | Association Group | Legend Index | Type of Dynamics |
|---|---|---|---|---|---|---|
| 7 | | Broad leaf—spruce forests (Bl-Sp) (*Quercus robur—Tilia cordata—Picea abies*) | 21 | Oak—linden—spruce broad herbs (*Carex pilosa, Galeobdolon luteum, Aegopodium podagraria, Asarum europaeum, Pulmonaria obscura, Ranunculus cassubicus, Stellaria nemorum*) | Bl-Sp_Bh | N |
| 8 | | Birch forests (B) (*Betula pendula, B. pubescens*) | 22 | Birch with spruce and aspen small herbs (*Oxalis acetosella, Pyrola rotundifolia, Luzula pilosa*) | B_Sh | LD |
| | | | 23 | Birch with spruce and aspen small herbs–broad herbs (*Oxalis acetosella, Athyrium filix-femina, Calamagrostis arundinacea, Rubus saxatilis, Galeobdolon luteum, Aegopodium podagraria, Pyrola rotundifolia, Cirriphyllum piliferum*) | B_ShBh | LD |
| | | | 24 | Birch with spruce and grey alder broad herb (*Aegopodium podagraria, Carex pilosa, Galeobdolon luteum, Pulmonaria obscura, Stellaria nemoru*) | B_Bh | LD |
| | | | 25 | Birch moist herb—broad herb (*Salix caprea, Filipendula ulmaria, Athyrium filix-femina, Urtica dioica, Calamagrostis arundinacea, Impatiens noli-tangere, Pulmonaria obscura, Geum rivale, Atrichum undulatum*) | B_MihBh | LD |
| | | | 26 | Birch with spruce and aspen grass-marsh (*Filipendula ulmaria, Calamagrostis canescens, Phragmites australis, Carex acuta, C. vesicaria, Scirpus sylvaticus, Aulacomnium palustre, Climacium dendroides*) | B_Gm | N |
| | | | 27 | Birch with spruce, aspen, and willow meadow herbs (*Bromopsis inermis, Calamagrostis arundinacea, C. epigeios, Fragaria vesca, Lysimachia nummularia, Veronica chamaedrys, Deschampsia cespitosa*) | B_Mh | LD |
| | | | 28 | Birch with spruce dwarf shrubs–herbal-sphagnum (*Chamaedaphne calyculata, Vaccinium uliginosum, Eriophorum vaginatum, Carex lasiocarpa, Sphagnum* spp., *Polytrichum commune*) | B_DshHSh | ShD |
| 9 | | Aspen forests (As) (*Populus tremula*) | 29 | Aspen with birch, spruce, oak, and linden broad herbs (*Corylus avellana, Aegopodium podagraria, Galeobdolon luteum, Carex pilosa, Mercurialis perennis*) | As_Bh | ShD |
| | | | 30 | Aspen with birch, spruce, oak, and bird cherry moist herbs—broad herbs (*Padus avium, Athyrium filix-femina, Crepis paludosa, Filipendula ulmaria, Urtica dioica, Pulmonaria obscura, Equisetum pratense, Stellaria nemorum, Impatiens noli-tangere, Atrichum undulatum, Plagiomnium cuspidatum*) | As_MihBh | ShD |
| 10 | | Grey alder forests (GAl) (*Alnus incana*) | 31 | Grey alder moist herbs—broad herbs (*Urtica dioica, Campanula latifolia, Filipendula ulmaria, Rubus idaeus, Aegopodium podagraria, Chrysosplenium alternifolium, Myosoton aquaticum, Stellaria nemorum, Plagiomnium undulatum*) | GAl_MihBh | N |
| 11 | | Black alder forests (BAl) (*Alnus glutinosa*) | 32 | Black alder moist herbs—broad herbs (*Impatiens noli-tangere, Urtica dioica, Milium effusum, Paris quadrifolia, Ranunculus cassubicus*) | BAl_MihBh | N |
| | | | 33 | Black alder grass-marsh (*Urtica dioica, Filipendula ulmaria, Phragmites australis, Carex appropinquata, C. vesicaria, Calla palustris, Humulus lupulus*) | BAl_Gm | N |

**Table A4.** Analysis of the size of the training sample.

| Ass. Gr. Number | Number of Points | Average Polygon Area, ha | Total Area of Polygons, ha | Number of Pixels for the Training Sample | Convergence Ass. Gr. by Test Sample | Percentage of Forest Area, % |
|---|---|---|---|---|---|---|
| 1 | 32 | 0.57 | 7.37 | 84 | 60 | 5.3 |
| 2 | 35 | 5.28 | 58.06 | 197 | 55 | 5.5 |
| 3 | 124 | 0.97 | 10.63 | 155 | 23 | 6.8 |
| 4 | 132 | 0.60 | 6.60 | 166 | 18 | 4.8 |
| 5 | 22 | 0.73 | 8.05 | 45 | 30 | 1.0 |
| 6 | 13 | 0.41 | 4.90 | 30 | 67 | 0.7 |
| 7 | 62 | 0.35 | 3.14 | 71 | 13 | 1.4 |
| 8 | 91 | 0.52 | 5.23 | 104 | 21 | 3.4 |
| 9 | 31 | 1.30 | 14.28 | 69 | 41 | 2.0 |
| 10 | 16 | 1.88 | 18.76 | 65 | 63 | 1.0 |
| 11 | 41 | 0.78 | 8.58 | 132 | 56 | 1.5 |
| 12 | 38 | 1.65 | 16.52 | 88 | 41 | 0.7 |
| 13 | 46 | 2.44 | 29.24 | 134 | 69 | 2.8 |
| 14 + 15 | 56 | 2.45 | 23.75 | 182 | 54 | 3.8 |
| 16 | 63 | 2.12 | 23.31 | 148 | 46 | 1.9 |
| 17 | 14 | 0.95 | 7.62 | 85 | 81 | 2.6 |
| 18 | 46 | 11.21 | 123.28 | 422 | 89 | 4.6 |
| 19 | 52 | 6.42 | 64.19 | 233 | 71 | 3.7 |
| 20 | 109 | 4.43 | 48.74 | 470 | 77 | 12.7 |
| 21 | 35 | 1.76 | 17.56 | 82 | 53 | 1.0 |
| 22 + 23 | 26 | 0.87 | 11.24 | 58 | 80 | 0.8 |
| 24 | 131 | 2.01 | 24.07 | 212 | 55 | 13.7 |
| 25 | 13 | 1.33 | 13.27 | 59 | 46 | 0.7 |
| 26 | 18 | 0.90 | 9.85 | 73 | 46 | 1.7 |
| 27 | 22 | 0.38 | 3.81 | 50 | 44 | 1.0 |
| 28 | 10 | 1.86 | 18.63 | 66 | 54 | 1.0 |
| 29 | 65 | 0.89 | 8.90 | 109 | 46 | 3.5 |
| 30 | 7 | 0.83 | 5.82 | 19 | 100 | 0.1 |
| 31 | 28 | 0.37 | 3.69 | 65 | 36 | 1.6 |
| 32 | 22 | 1.21 | 12.06 | 55 | 71 | 2.0 |
| 33 | 31 | 0.98 | 10.81 | 68 | 77 | 6.6 |
| 34 | 26 | 2.59 | 25.86 | 102 | 72 | * |
| 35 | 3 | 0.78 | 1.55 | 7 | 50 | * |
| 36 | 8 | 2.26 | 13.56 | 47 | 93 | * |
| 37 | 10 | 2.27 | 4.54 | 29 | 50 | * |
| 38 | 6 | 2.31 | 3.98 | 30 | 67 | * |

\* Areas of non-forested and unforested areas are not given in this table, because analysis and comparison are focused only on forested areas.

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
