# Peer review of "Environmental Performance of Regional Protected Area Network: Typological Diversity and Fragmentation of Forests"

_remotesensing, doi:10.3390/rs15010276_

Round 1

Reviewer 1 Report

This interesting article deals with the topic of forest fragmentation in the protected areas of the Russian Moscow region and shows a comparison of the indices already known in the literature for its quantification. The topic of this manuscript falls within the objectives of the journal. The topic is well presented and the English style and grammar are good, as the text is easy to follow. However, further efforts are needed to make better use of the work. The following is recommended to the authors.

Material and methods 

It should be better explained how agriculture and urbanised areas interfere with fragmentation. the work would deserve a comparison of the variation of the various indicators over a fairly long time span, at least 5 to 10 years.

Author Response

Thank you very much for your careful reading of the manuscript and valuable comments.

Our geobotanical study in general as well as this current study is focused on forest cover. It would be interesting and productive to include non-forest geobotanical studies in further studies. However, we have tried to improve the text with the addition of how agriculture and urbanised areas affected the structure of forest cover in the Moscow region. An insert with a historical digression was placed in the introduction. This certainly improved the understanding of the picture of green infrastructure development around the largest agro-industrial complex in the eastern part of Europe. 3 additional literary references were also given.

Current study has spatial context. Identification of the dynamics of indicators over the past 5-10 years, from our point of view, is an additional task that the authors did not set in this work.

Reviewer 2 Report

The article is interesting and the topic is worthy of investigation.  However, the wording is a bit confusing at times and the methodology does not present a clear correspondence with the presentation of results.  Consequently, it is recommended to incorporate a series of improvements.  The following are the issues that should be addressed by the authors:

METHODOLOGY

The explanation of the methodology section excessively descriptive, and does not detail sufficiently how it is addressed in each of the stages what is expected to appear in the results section.  This makes the process difficult for other researchers to reproduce, since the way to get the results is a kind of black box.  In addition, the whole is also a bit confusing, so it is recommended to incorporate a summary scheme of the methodological process, graphically detailing the different stages and expected intermediate results.

RESULTS

There is no clear correspondence between the structure proposed in the methodology section and the content of the results.  It is recommended to clarify this issue so that readers can clearly identify the methodological framework in the presentation of results.

DISCUSSION

In the scientific discussion section, a greater content of self-critical reflection on the part of the authors regarding their work is lacking, underlining the limitations of the research carried out, and detailing what issues should be improved in future lines of research.

Author Response

The authors are very grateful for the high assessment of our work, valuable comments and suggestions. We have significantly reworked the wording and improved use of terms.

Reviewer 3 Report

The study is quite interesting and set up a good and well-documented context on the importance of protected areas for biodiversity conservation and ecosystems functioning. Addressing the role of this kind of infrastructure for urban areas, from the remote sensing perspective show a highly valuable approach. In addition, the study show a sound and comprehensive methodological framework, which is also well described.

Specific comments

Figure 4, please provide legends for X-Y axis. In addition, I suggest a standardized scale for Y-axis, to avoid a misleading comparison.

Figure 5, I suggest the use initials of the natural derived communities (X-axis) instead of numbers.

Table 3, does not looks like a typical ANOVA table, please explain more on the comparison you are making and how, (Avg. Number of typological diversity by PA categories), i.e. it is not clear which PA categories differs (significance level) with others in terms of number of typological units.  

Lines 358-359, incomplete sentence, missing words, please review.

Discussion

This is a weak part of the paper. Most of the section are rephrasing of the results (i.e. lines 437-445; 448-449; etc.), besides the results of the study are not compared or contrasted with the literature (only a few references, and not very substantial, in the whole section), this is an important shortcoming, that the authors must reconsider. The Reference section contains enough literature that might be recalled to enrich the discussion. 

Some sentences from the discussion deserve a deeper analysis, i.e. in lines 472-477, what are the implications of this result in the efficiency of the PAs? To what extent this result affects the provision of some environmental services for the MR?. In addition, check lines 486-488; 494-495, 509-510, which does not provide insights in the discussion of the results, seem speculative and needs further literature support to make a consistent discussion.  As a result, the conclusion section does not show clearly the contribution of the study in term of the objective setting. Considering the important results of the study, I believe the Discussion section must be improved to strength the paper contribution.

Author Response

The authors thank the reviewer for the high appreciation of our work and the importance of the problem being solved. Your Specific comments are very valuable to us, all the right to the corresponding drawings will be done.

Reviewer 4 Report

The article is well developed and interesting. It seems to me that the results are extensively explained, although in Figure 6 there is no clear correlation, so it would be convenient to work with more data or have other relationships that describe the data.

Author Response

The authors are very grateful for the high assessment of our work.
